# Exploring the Link between Street Layout Centrality and Walkability for Sustainable Tourism in Historical Urban Areas

**Mustafa Aziz Amen** [1,*] , **Ahmad Afara** [1] **and Hourakhsh Ahmad Nia** [2]

[1] Design Department, The American University of Kurdistan, Duhok 1063, Iraq; ahmad.afara@auk.edu.krd
[2] Department of Architecture, Alanya University, Alanya 07400, Turkey; hourakhsh.ahmadnia@alanyauniversity.edu.tr
*  Correspondence: mustafa.amen@auk.edu.krd

**Abstract:** Walkability is considered a vital component of the urban configuration; urban spaces should promote pedestrian walking, which is healthier and increases social sustainability by connecting people in urban spaces. This article aims to find the link between the street layout centrality values and the people's walkability for sustainable tourism in historic areas. Moreover, it attempts to explore the linkage between the urban layout and visiting historical spaces in the urban layout. The approach to the research has two phases; the first is to find people density (the tourist density) in the historical areas, and the second is to measure the centrality values of the urban layout utilizing the spatial design network analysis tool (sDNA). The research found that the street network considerably impacts the final tourist distribution, mainly because of the betweenness centrality; consequently, spaces with low betweenness centrality values are less reachable by the tourists in the historical area, although it has a high closeness centrality. The research concluded that considering the street network is necessary concerning the tourists' walkability since it affects their density in the urban layout.

**Keywords:** centrality values; sDNA; historical places; street network; walkability





## 1. Introduction

Urban sustainability is the process of developing a built environment that meets people's needs while avoiding inappropriate social or environmental impacts; therefore, the concept of sustainability is used to analyze the evolution of the commercial fabric [1], rehabilitation challenges [2,3], analyze the urban green infrastructures in the climate change fight [4], conscious consumer behavior [5], Recycling and Source Separation Practices [6], spatial perception [7] and air quality [8].

The core of urban sustainability concerns issues about the standards of living enjoyed by humans throughout their lifetimes [9]. Accordingly, it is a complex, multidimensional concept that needs a set of sustainability indicators rather than a single hand to measure it [10]. As a sequence, the primary concerns of the idea of sustainability are the preservation of the natural environment [11,12], the thermal effect of ecological network factors [13], tourism flow [14], accessibility and economic linkage [15–17], hotspots during the COVID-19 pandemic [18], Urban Intelligence for Carbon Neutral Cities [19], the complex structural to deal with the disturbance of emergencies [20], and the walkability [21–29]. And to create vibrant and livable cities that support walkability and sustainability, an effort to promote active street network connections must be made with shorter block lengths and many intersections to facilitate more direct travel between locations [30,31]. The study aimed to provide a more coherent presentation of the problem, objectives, and research gap by investigating the relationship between street layout centrality, walkability, and sustainable urban tourism in historical areas and examining the influence of centrality values of urban layouts, specifically betweenness and closeness centrality, on walkability and tourist distribution. The research gap lies in need for a deeper understanding of how

the configuration of street networks in historical urban areas affects sustainable tourism by shaping tourists' walking patterns and visiting preferences. To refine the studies objectives, the authors focused on: (1) exploring the impact of the street network's centrality values on tourist distribution, with particular emphasis on betweenness centrality, (2) analyzing the role of street layout in promoting walkability in historical urban areas, and (3) providing recommendations for urban planners and policymakers to optimize street network design for sustainable tourism. By addressing these objectives, this study seeks to contribute to the existing knowledge on the interplay between urban layouts, walkability, and sustainable tourism in historical areas, ultimately offering valuable insights for developing more accessible and sustainable urban spaces.

## 2. Theoretical Background

The concept of walkability is increasingly essential in theory and practice, significantly affecting urban sustainability. Through research and implementation, walkability has matured into a more comprehensive, organic, multi-dimensional description of the relationships and dynamics between pedestrians, urban space, and the social practices of utilization [32–34].

Street layout has a crucial impact on the walkability of people exploring the urban layout [21,22,35,36], where scholars studied the relationship between walkability and tourists' perception [31], obesity [37,38], building block [39], tourist and wellbeing [35], Promoting Sustainable Urban Neighborhood [40], Iconic Heritage Destination [41,42], Sustainable Tourism Impact on Residents [43] and Geographic Distribution of High Body Mass [44]. However, the relationship between the walkability for sustainable tourism and the street network centrality indicators in the historic areas is unclear, particularly the impact of specific centrality indicators on guiding people in the historical area and, more specifically, the relationships between the tourist density and the street network centrality. Although the historical sites are located very close to each other and are well-defined in the tourist guide map, the authors attempted to explore and find answers to why tourists visit some historic sites and ignore others. And to approach answering their question, the authors believe there is a need to find the central locations in the urban layout, which is the most critical factor in the street network that plays a significant role in connecting the street network [45–47]. It is worth mentioning that the central locations have played an essential role in studying land use [48], crime locations [49,50], urban planning [51–54], urban economy [55,56], architectural analysis [57], traffic congestions [58].

Knowing that street networks are complex to study, determining their central values simplifies the process, such as studying their closeness, farness, and betweenness. As a result, the first step in exploring networks is to convert the street network to a graph [59–61]. Following that, there are two methods to study the graph and find the centrality values; the first one is the primal approach that uses the actual network [62–64], while the second one is the dual approach representing in space Syntax [65–70].

The research study adopted the primal mode as defined by the Spatial Design Network Analysis (sDNA) [71,72]. The sDAN is used to find two centrality values in the spatial network; it computes measures of accessibility (reach, mean distance/closeness centrality, gravity), flow (bidirectional betweenness centrality), and efficiency (circuity) as well as convex hull properties, localized within lower and upper-bounded radial bands [71]. Therefore, the paper used the tool to analyze the spatial network's closeness, betweenness, and efficiency with the spatial walkability of the urban layout through tourist density.

## 3. Materials and Methods

The authors adopted two steps to find the relationship between the tourist destinations and the street network centrality in the spatial layout. Manual counts were conducted at intersections to find the tourists' density by recording short videos at the case study intersections. Therefore, they used a manual account to extrapolate pedestrians' weekly volumes [73]. Zhou et al. used recorded video to study the impacts of mobile phone

distractions on pedestrian crossing behavior at signalized intersections [74]. In contrast some other scholars used short videos for counting, behavior, and safety analysis at intersections [75]. These short videos were used to study the bicycle and pedestrian counts at signalized intersections [76].

This study focuses on the relationship between street layout centrality and walkability in the context of sustainable tourism in historic urban areas. A crucial aspect to address is the description of the study area from which data was collected. The research was conducted in the Turkish part of Nicosia's old city; the Turkish part of Nicosia's old city is a beautiful and historically rich area that offers a unique blend of cultures, traditions, and architectural styles. As the northern section of the divided capital of Cyprus, this part of Nicosia is characterized by its narrow, winding streets, traditional stone houses, and vibrant markets. The Selimiye Mosque, formerly known as the Cathedral of Saint Sophia, is a testament to the city's diverse history, showcasing a fascinating fusion of Gothic and Ottoman architectural elements. The ancient Venetian walls encircle the old city and add to its charm and historical allure. They invite visitors to explore the rich tapestry of cultures shaping this captivating part of Nicosia (Figure 1a). accordingly, 32 students from Girne American University were placed at central street intersections to determine tourist density. This area was chosen due to its rich historical significance, featuring numerous administrative offices, organizations, historical sites, and tourist attractions that expect visitors from all over Cyprus. In this regard, the primary locations for measurements included The Buyuk Han, Bedesten, and the area around the Venetian Column, among the most important tourist destinations.

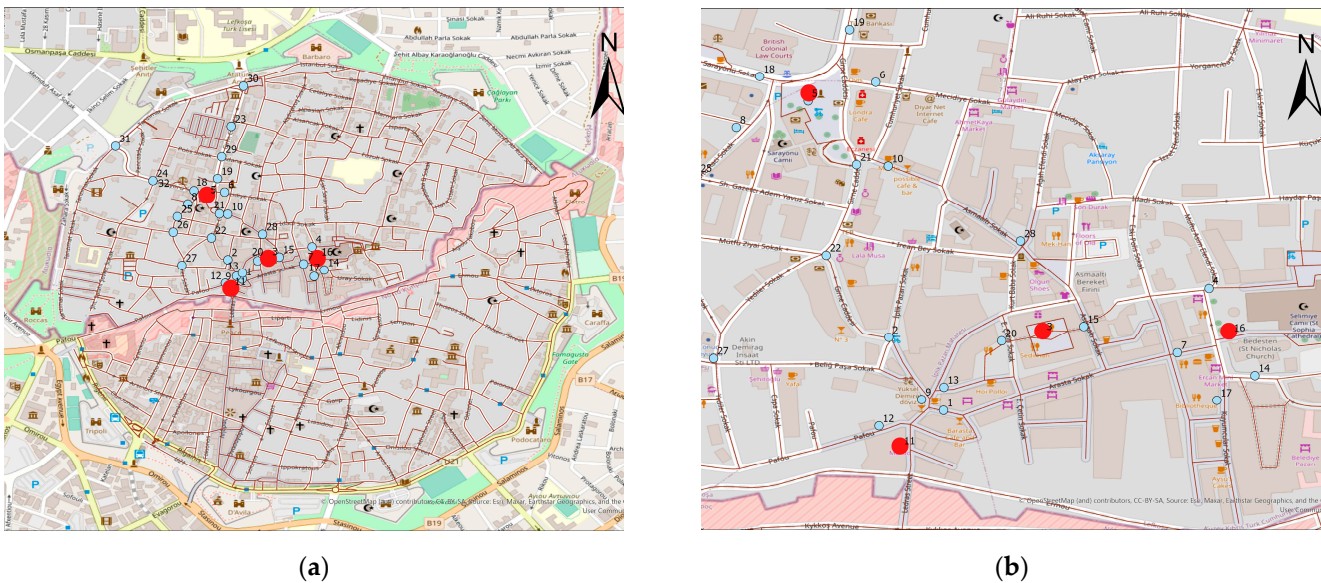

(a)　　　　　　　　　　　　　　　　　　　　　　　　　　　　　(b)

**Figure 1.** (**a**) students' location in Nicosia city; (**b**) location of the historical building and monuments that tourists are expected to visit.

This study has provided significant insights into the connection between street layout centrality and walkability in historical urban areas, mainly focusing on sustainable tourism. However, it's crucial to acknowledge that these findings are derived from a relatively limited data set, mainly from the Turkish part of Nicosia's old city. This focused scope presents an inherent limitation, as the extrapolation of these results to broader or differing contexts may be challenging due to the studied area's unique sociocultural and geographical characteristics.

With that in mind, the necessity for additional data becomes apparent. Conducting research in diverse historical urban areas across different regions or countries could help create a more comprehensive understanding of how street layout centrality influences walkability. Further, performing more robust analyses using advanced statistical methods

is pivotal, ideally integrating different data sources such as pedestrian behaviors, urban layout characteristics, and tourism patterns. This would not only strengthen the validity of the findings but also aid in discerning complex relationships and patterns. To generalize the study's findings, a more extensive and diversified data collection is required, considering the various factors that can influence walkability in different historical urban settings.

The attractiveness and importance of the study area lie in its historical elements, such as well-defined monuments and architecture. These elements are vital for understanding the context of sustainable urban tourism, as they provide insights into the importance of the area's tourist destinations. Facilities in the area that contribute to its walkability include local restaurants, coffee shops, gift, and craft shops, live music venues, and local festivals, particularly in the Buyuk Han. These amenities attract tourists and contribute to the area's density scores, highlighting the importance of striking a balance between historical preservation and tourism development in historic areas. Considering this fact, constraints in the area include the varying betweenness and closeness centrality values of different locations, which affect tourist distribution and walkability. The study found that betweenness centrality had a higher impact on tourist distribution than closeness centrality, suggesting that tourists frequently visit places with high betweenness centrality. Therefore, understanding the constraints of the area's street network and its impact on walkability is essential for urban planners and policymakers when designing sustainable and accessible urban spaces in historic urban areas.

In the study, the authors used the terms INNs and OUTs to represent the number of people arriving at an intersection (INN) and the number of people leaving the intersection (OUT). By analyzing the INNs and OUTs at various street intersections in the historical urban area, the authors aimed to understand pedestrian movement patterns, walkability, and people's propensity to visit attractions by walking.

The INNs and OUTs can provide insights into the level of walkability in the area and the factors that might influence people's decisions to walk to certain attractions. For instance, more INNs could indicate that an intersection is a popular destination or a well-connected point in the urban layout. In comparison, more OUTs suggest that people leave that area to explore other attractions or return to their origin points. Additionally, the balance between INNs and OUTs at specific intersections might help understand pedestrian traffic's overall flow and distribution. However, it is essential to analyze the INNs and OUTs data in conjunction with street layout centrality values, such as closeness, betweenness, and farness, to gain a more comprehensive understanding of the factors influencing walkability and people's propensity to visit attractions by walking.

The authors distributed 32 students at the city's street intersections Figure 1a to find the people's walking destinations. The students were equipped with a camera and were required to record a short video for each street intersection, accounting for people passing the intersection while coming to the intersection, the INN, or going far from the junction, the OUT. The videos were recorded three times a day, with a limited recording duration of 15 min, and at specific day times, and each video record starts at 9:00 a.m., 12:00 p.m., and 16:00.

The article's authors collected data over two days, one on a working day and the other on a weekend, to observe the differentiation in the number of people visiting the area on different occasions. By choosing two distinct types of days, they aimed to obtain a snapshot of pedestrian activity in the historical area during regular working days and weekends, which could show varying patterns of walkability and tourist density. However, it is essential to note that collecting data from only two days might only partially represent the actual situation. While it offers a glimpse into the area's walkability patterns and tourist density, the limited data collection period may need to account for seasonal variations, special events, or other factors that could influence pedestrian activity over time. A more comprehensive study would require data collection over a more extended period, including different seasons and various weekdays, to provide a more accurate and representative analysis of the link between street layout centrality and walkability in historic urban areas.

Since the city features several administrative offices, organizations, historic sites, and tourist attractions that expect visitors from all over Cyprus, the students were divided between the primary and secondary portals leading to the major historical sites. The primary location for measurements was The Buyuk Han (Point 3), Bedesten (Point 16), and the area around the Venetian Column are the three most important tourist destinations (Point 5). However, it is essential to note that most individuals who visit these areas from the lower portion of the city must pass through the Checkpoint (Point 11) region, which was once predicted to have the greatest population density. Figure 1b.

The students were instructed to manually count the number of individuals shown in the short videos they recorded to determine how many people arrived and exited the junction. Having calculated the number of people they recorded, the students transformed the data into GIS MapIt, a free application available in the Android Market. Later, the data were combined as one general map in Carto DB.

The video recording process repeated for two days. The first was a regular working day, Monday, and the second was a weekend, Sunday. In addition to the intersection, the authors chose four locations in the case study as Stationary Points; those points are synonymous with the tourists' urban destination; accordingly, two students in each place recorded a short video and then counted the number of people who were there in the selected timeframe set to the study. The authors then inspected and studied the data collected on the chosen days, Monday and Sunday.

Since the data were nonparametric, the study adopted the Freidman two ANOVA to check the assumptions; this step is significant in deciding whether to adopt both measurements or withdraw one of the tests used to examine the differences in conditions when there are more than two conditions, and the same entities have provided scores in all states (so, each case contributes several scores to the data), and when we want to counteract the presence of unusual circumstances, or we have violated one of the assumptions from

$$F_r = \left[\frac{12}{Nk(k+1)}\sum_{i=k}^{k} R_i^2\right] - 3N(k+1) \tag{1}$$

$R_i$ is the sum of ranks for each group, $N$ is the total sample size, and $k$ is the number of conditions [77,78].

After measuring the people density in the studied space, the research also adopted the spatial design network to measure the centrality criteria to find the central location in the city precisely to measure the closeness in terms of network quantity penalized by distance and betweenness. While the last step, and to answer the paper's main question, the authors use the Spatial Pearson correlation method to explore the correlation between the density of the people and the spatial layout centrality indicators through spatial design network analysis to measure the farness, betweenness, and efficiency (circuity) [71,79].

*3.1. Closeness, Farness, and Network Quantity Penalized by Distance*

3.1.1. Closeness

Closeness refers to the average distance from a specific point (node) to all other points (nodes) within a network. It measures how easily accessible or well-connected a location is within the network. In the context of urban street layouts, closeness can provide insights into the walkability and efficiency of an area. A higher closeness value indicates shorter average distances to other locations, suggesting better connectivity and accessibility. Closeness is the measuring potential for "to-movement" [61,80–82]. The indicators measure the number of lines in the shortest path between an actor and other actors in the network [83–89].

sDNA doesn't measure closeness; it measures farness, which tells the same thing differently. The literature often defines closeness as 1/farness, though this has an exponential distribution, so statistically, it is harder to work with an alternative definition of closeness that doesn't suffer from this problem is −farness, as clarified in Equation (2) [71,72,79].

### 3.1.2. Farness

Farness is the inverse of closeness and represents the total distance from a specific node to all other nodes within the network. In an urban context, farness can be used to identify relatively isolated or distant areas from the rest of the network. High farness values suggest a location is less accessible or less well-connected, which may impact the area's economic and social vitality.

$$\text{Farness}(x) = \frac{\sum_{y \in Rx} d_m(x, y) W(y) P(y)}{\sum_{y \in Rx} W(y) P(y)} \tag{2}$$

where

- The set of polylines in the network radius from link x is denoted Rx
- The distance, according to a metric M, along a geodesic defined by M, between an origin polyline x and a destination polyline y is denoted $d_m(x,y)$
- The weight of a polyline y is denoted W(y).
- The proportion of any polyline y within the radius is denoted P(y). In discrete space analysis, this always equals 0 or 1, i.e., $y \in Rx \Leftrightarrow P(y) = 1$. In continuous space, $0 \leq P(y) \leq 1$ [71,72]

The research adopted NQPD as an alternative for the farness closeness that considers both quantity and accessibility of network weight. Typically, Farness considers only accessibility. As a result, the closeness measured according to Equation (3), Network quantity penalized by distance (gravity model) NQPD, is a form of closeness, commonly referred to as a gravity model, that considers both the amount and the accessibility of the network weight. By contrast, Farness only considers accessibility, as clarified below.

$$\text{NQPD}(x) = \sum \frac{W(y)P(y)^{nqpdn}}{d_m(x,y)^{nqpdn}} \tag{3}$$

NQPD default is set to 1 but can be set to other values in advanced config (they stand for NQPD numerator and denominator, respectively). The problem for any given application is determining the correct values for each, i.e., the relative importance of network quantity and accessibility [71,72]

### 3.2. Betweenness

Betweenness measures the importance of a node or an edge within a network based on the number of shortest paths that pass through it. In an urban street layout, betweenness can be used to identify critical junctions, bridges, or streets that serve as essential connectors between different areas. High betweenness values indicate that a particular node or edge is crucial for maintaining connectivity within the network, and any disruption to it may have significant consequences on the overall network efficiency. Betweenness centrality is generally regarded as a measure of others' dependence on a given node and, therefore, as a measure of potential control [85,90,91], betweenness counts the number of geodesic paths [62,83,86] that pass through a vertex, i.e., the number of times the vertex lies on the shortest path between other pairs of vertices [63,71]. The betweenness centrality (BC) is vital for understanding the structure of large complex networks [86] sDNA clarifies the betweenness as below.

$$\text{Betweenness}(x) = \sum_{y \in N} \sum_{z \in R_y} W(y)W(z)P(z)OD(x,y,z) \tag{4}$$

where

$$OD(x,y,z) = \begin{pmatrix} 1, & \text{if xison the first geodesic found from ytoz} \\ \dfrac{1}{2}, & if\ x\ =\ y\ \neq\ z \\ \dfrac{1}{2}, & if\ x\ =\ z\ \neq\ y \\ \dfrac{1}{3}, & if\ x\ =\ z\ =\ y \\ 0, & otherwise \end{pmatrix}$$

Urban betweenness is used in urban layout to assess the significance of a node in terms of transport flows [64]. We use the betweenness centrality, which is the number of shortest paths, gi, between two other nodes that pass through the node [63]. So, the measure is used to measure the people's flow of information.

Implementing a threshold for closeness, betweenness, and fairness can augment the analysis of street layout centrality and its impact on walkability, especially for sustainable tourism in historic urban areas. In this study, a threshold refers to a critical value or range of values for these centrality measures, beyond which the degree of walkability or tourist distribution might significantly change. In this case, thresholds can serve as indicators to delineate well-connected and less-connected areas, aiding urban planners and policymakers in optimizing street network design and improving pedestrian accessibility.

In the study, closeness refers to the average distance from a specific point (node) to all other points (nodes) within a network, indicating how easily accessible or well-connected a location is. A defined closeness threshold can serve as a benchmark to determine whether a particular area within the historic urban zone is "close enough" to be accessible or attractive to tourists for walking. Similarly, betweenness represents the importance of a node or an edge within a network based on the number of shortest paths passing through it. A betweenness threshold could help identify critical nodes that serve as essential connectors for tourist distribution. Lastly, fairness, the inverse of closeness, represents the total distance from a specific node to all other nodes within the network. A farness threshold can identify areas that might be too isolated or distant from the rest of the network, potentially deterring tourist walkability. Utilizing these thresholds would allow a more nuanced understanding of the influence of street layout centrality on walkability, ultimately contributing to sustainable tourism in historic urban areas.

### 3.3. Moran Index

Finding the spatial autocorrelation is crucial to understand the data distribution spatially; the theory of spatial autocorrelation has been a critical element of geographical analysis for more than twenty years [92,93] autocorrelation is essential to find out the correlations [94–96] for the closeness and betweenness; high autocorrelation indicates that most people's density might be connected to the high betweenness and closeness (NQPD) using the equation below.

$$I = n \frac{n}{s_o} \frac{\sum_{i=1}^{n} \sum_{j=1}^{n} w_{i,j} z_i z_j}{\sum_{i=1}^{n} z_i^2} \tag{5}$$

where:

$Z_i$ is the deviation of an attribute for a feature is from its mean.

$w_{i,j}$ is the spatial weight between components i and j.

Figure 2 below clarifies the research methodology, composed of five steps.

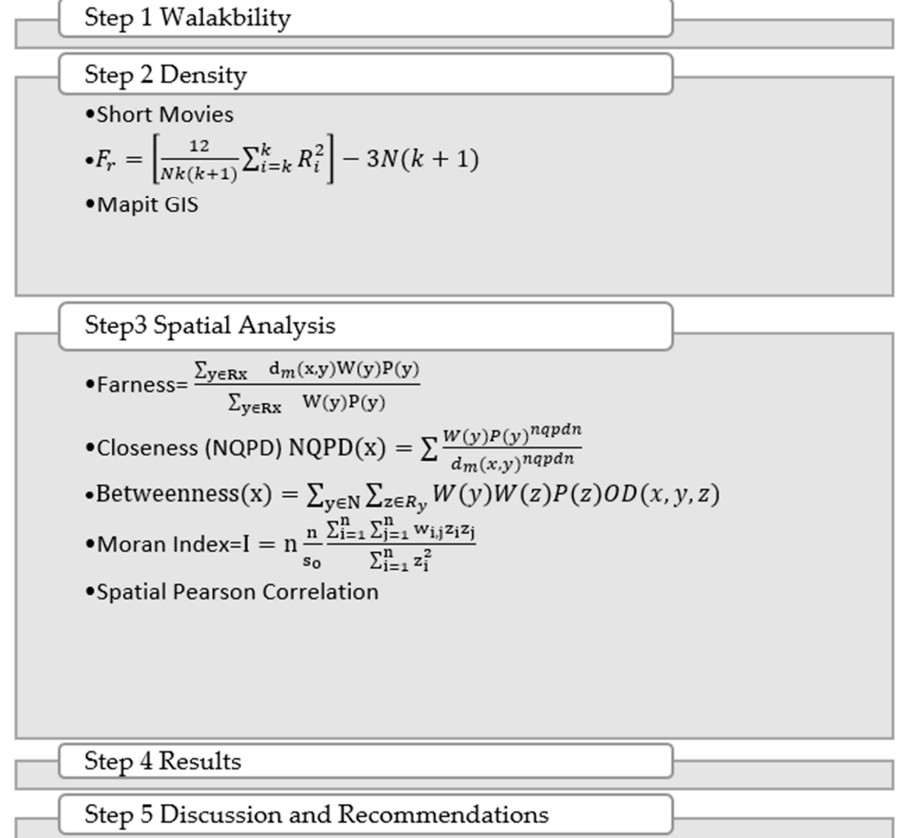

**Figure 2.** The research methodology.

## 4. Results

The article establishes the link between street layout centrality and walkability for sustainable tourism in historical urban areas using centrality measures like closeness, betweenness, and farness. By analyzing these measures, the authors can identify well-connected, accessible, and efficient street networks crucial for promoting walkability in historic urban areas. The observed link suggests that areas with higher closeness and betweenness values, and lower farness values, are more conducive to walking and exploration by tourists. These well-connected networks facilitate easy navigation and access to various points of interest, promoting sustainable tourism practices that reduce dependence on vehicular transportation. Consequently, this enhances the overall experience for tourists and fosters a more immersive engagement with the local culture, history, and environment. Regarding the implications of this link, it needs to highlight its significance for urban planning and heritage management, particularly in historic urban areas. Planners and policymakers can use these centrality measures to prioritize interventions that enhance walkability, such as pedestrianization, improved signage, and the development of public spaces. By fostering walkable environments, cities can promote sustainable tourism practices that benefit both visitors and local communities, reduce negative environmental impacts, and contribute to the long-term preservation of historical urban areas.

According to the initial maps of the working day on the 30 May, the map shows a significant variation in the number of people density in the 32 sites the researchers selected to measure. The outcome of the working day reveals that most of the intensity is placed in the Bedesten area (point 16) and the lowest density in the Venetian Column (Point 5), as clarified in Figure 3a,b. However, according to the survey done on the day off, 29 May, the result reveals that most of the intensity happens in the Buyuk Han (point 11), and the lowest density happens in the Venetian Column (Point 5), as clarified in Figure 3c,d. Also,

it is worth mentioning that the links and the intersections that connect the major historical area show low values in terms of people density; those points are 21, 22 and 27, respectively.

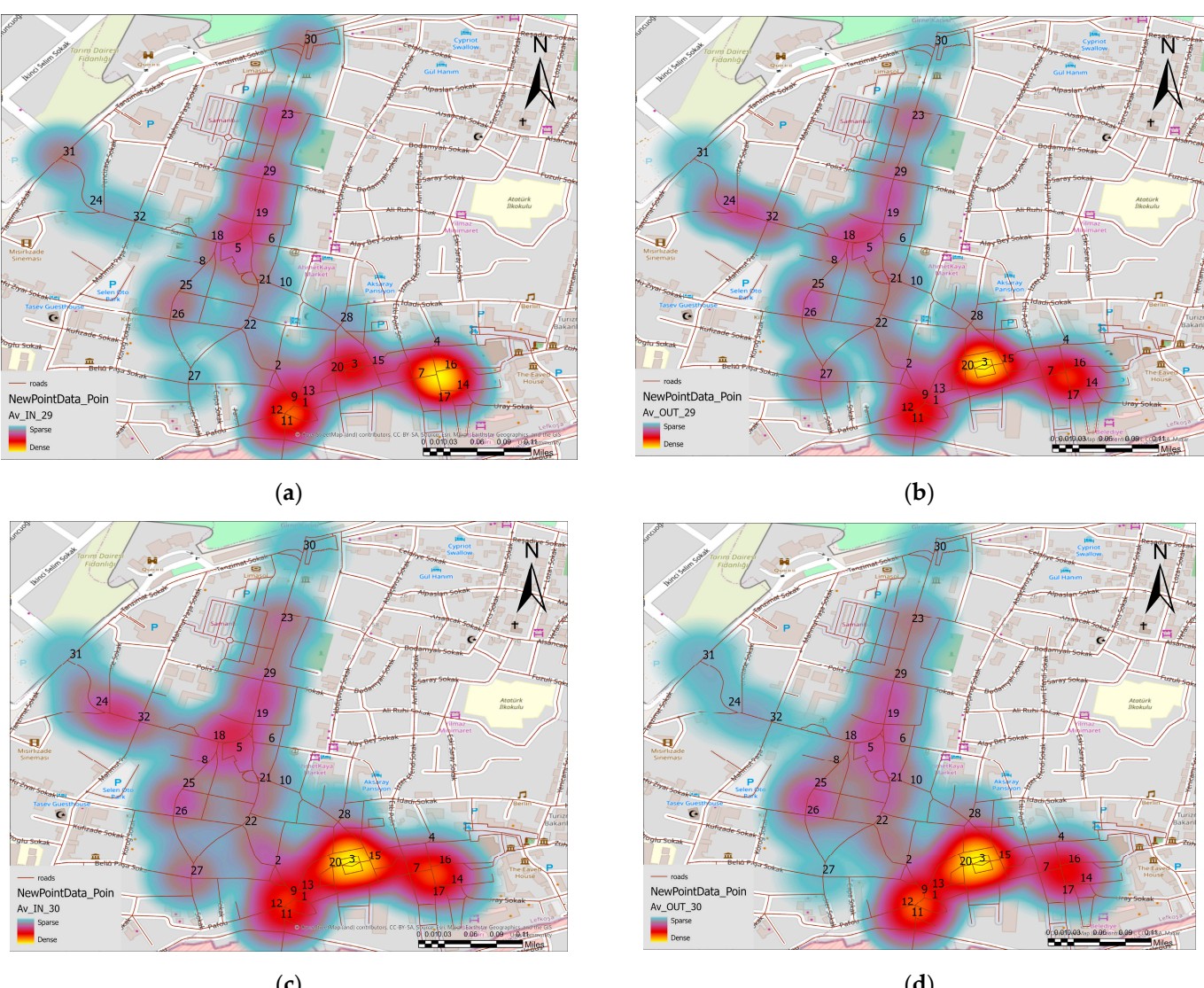

**Figure 3.** (**a**) Average people entering the locations on off day; (**b**) Average people leaving the locations on off day; (**c**)Average people entering the locations on Working day; (**d**) Average people leaving the locations on Working day.

After determining the population density, the study employed Spatial Design Network Analysis (sDNA) to determine the spatial centrality of the urban plan, namely the closeness (NQPD) and the betweenness. As a result, the closeness analysis of the spatial layout utilized a 500-m radius and segment lengths as weights; consequently, the centrality values for the spatial layout ranged from the lowest values, which were located between (3.00–69.67), to the highest values, located between (70.01–100.01). (105.18–126.67) as it clarified in Figure 4. Regarding the centrality values for the historical places, there is an extensive range of proximity values between the four locations, as clarified in Figure 3a. Accordingly, the closeness centrality scores of the four locations ranged between (105.18 and 126.67). The street segments that connect points 11, 3, 5, and 16 have relative values regarding closeness centrality, indicating that people or tourists have an equal opportunity to visit these locations regarding closeness centrality.

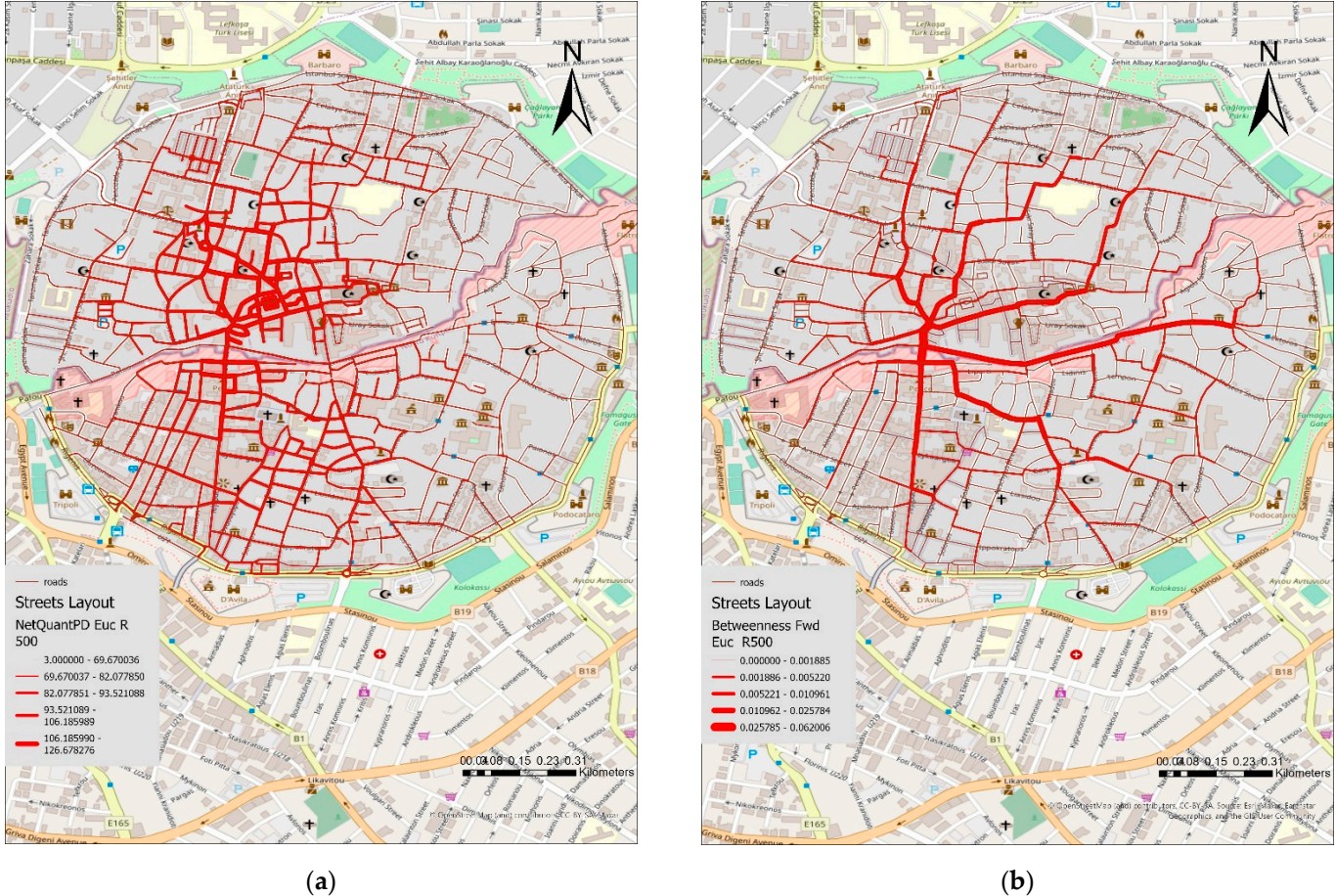

**Figure 4.** (**a**) Coleness Centrality measured through NQPD; (**b**) Betweenness centrality.

However, by looking at the Betweenness centrality or the shortest path crossing every intersection in the street, the layout is held within a 500 m radius and considers the segments as a weight. The betweenness centrality values ranged from (0.000–0.001) as the lowest values to the highest values (0.025–0.067. The low values represent segments or paths in the spatial layout with fewer short paths crossing them, while the high values represent paths with an increased number of short paths crossing them accordingly.

In consequence, the high betweenness values located in the center of the old city show a segregation between the betweenness centrality values connecting points 3, 5, 11, and 16. Thus, there are three equal streets in terms of betweenness coming from Check Point (point 11) and distributed simultaneously to the other locations in the spatial layout. The results of the density and the centrality values are clarified in Table 1. The Venetian as a measured location has the lowest betweenness value compared to the other three stationary locations, as described in Table 1 below.

The authors used Moran Index to find the spatial autocorrelation among the closeness and betweenness data; the results show that both values are randomly distributed and indicate that they are not clustered, closeness (NQPD) spatial autocorrelation Moran's index is 0.0526, Z-score is 0.970, and the *p*-value is 0.331, while betweenness spatial autocorrelation Moran's Index is 0.097, z-score 1.385 and the *p*-value is 0.166 which suggests that there is a broken link between the street network segments.

**Table 1.** People Density in the working day 29th and 30th of May.

| Point | In-29 | Out-29 | In -30 | Out-30 | AV-29 | AV-30 | NQPD | Betweenness |
|---|---|---|---|---|---|---|---|---|
| 1 | 49 | 41 | 46 | 36 | 45 | 41 | 102.91 | 0.00609 |
| 2 | 38 | 30 | 55 | 35 | 34 | 45 | 105.09 | 0.01688 |
| 3 | 57 | 58 | 130 | 129 | 58 | 130 | 114.34 | 0.002256 |
| 4 | 11 | 14 | 8 | 11 | 13 | 10 | 108.80 | 0.00247 |
| 5 | 36 | 37 | 25 | 24 | 37 | 24 | 109.56 | 0.009457 |
| 6 | 5 | 12 | 3 | 12 | 8 | 8 | 106.55 | 0.003752 |
| 7 | 58 | 16 | 62 | 35 | 37 | 49 | 107.77 | 0.014994 |
| 8 | 3 | 6 | 4 | 5 | 4 | 5 | 94.45 | 0.0011 |
| 9 | 11 | 12 | 14 | 13 | 11 | 14 | 126.68 | 0.042536 |
| 10 | 5 | 4 | 8 | 7 | 5 | 8 | 108.12 | 0.001334 |
| 11 | 73 | 70 | 75 | 92 | 72 | 84 | 119.30 | 0.062 |
| 12 | 3 | 2 | 11 | 7 | 3 | 9 | 108.76 | 0.006631 |
| 13 | 10 | 5 | 9 | 15 | 8 | 12 | 117.09 | 0.0031 |
| 14 | 56 | 35 | 58 | 47 | 46 | 52 | 96.23 | 0.001774 |
| 15 | 27 | 26 | 57 | 31 | 27 | 44 | 119.10 | 0.002357 |
| 16 | 78 | 75 | 48 | 47 | 77 | 48 | 99.45 | 0.012631 |
| 17 | 7 | 4 | 8 | 4 | 6 | 6 | 98.26 | 0.000224 |
| 18 | 17 | 9 | 56 | 19 | 13 | 38 | 92.78 | 0.005063 |
| 19 | 53 | 44 | 54 | 45 | 49 | 49 | 100.41 | 0.008649 |
| 20 | 22 | 17 | 27 | 37 | 20 | 32 | 114.92 | 0.002851 |
| 21 | 26 | 24 | 40 | 41 | 25 | 41 | 159.39 | 0.006979 |
| 22 | 15 | 21 | 39 | 44 | 18 | 42 | 105.09 | 0.01688 |
| 23 | 53 | 44 | 54 | 45 | 49 | 49 | 92.52 | 0.005057 |
| 24 | 17 | 9 | 56 | 19 | 13 | 38 | 71.95 | 0.000726 |
| 25 | 24 | 23 | 40 | 43 | 24 | 42 | 94.45 | 0.0011 |
| 26 | 26 | 22 | 39 | 41 | 24 | 40 | 98.95 | 0.002184 |
| 27 | 17 | 9 | 56 | 19 | 13 | 38 | 102.20 | 0.002184 |
| 28 | 26 | 22 | 40 | 41 | 24 | 41 | 0.00 | 0.014842 |
| 29 | 53 | 44 | 54 | 45 | 49 | 49 | 97.21 | 0.0059 |
| 30 | 37 | 36 | 25 | 27 | 37 | 26 | 89.10 | 0.002795 |
| 31 | 36 | 38 | 24 | 24 | 37 | 24 | 68.96 | 0.000568 |
| 32 | 17 | 9 | 56 | 19 | 13 | 38 | 85.63 | 0.003185 |

## 5. Discussion

The research gap bridged in this study lies in examining the relationship between street layout centrality, walkability, and tourist density in historic urban areas. The study establishes a correlation between these factors by using centrality measures (closeness and betweenness) and people density data collected from various locations in the old city of Nicosia This understanding helps identify the significance of street networks and their impact on tourist service facilities and historical preservation. Therefore, the implications of this study are valuable for urban planning and heritage management in historic urban areas. The findings reveal that areas with higher centrality values, in terms of closeness and betweenness, are more likely to experience higher tourist density. Additionally, local restaurants, coffee shops, and gift and craft shops are vital in attracting tourists to specific locations. This highlights the importance of balancing historic preservation with providing tourist services and amenities. Moreover, the study emphasizes the need to consider the effect of street networks on establishing tourist service facilities. By improving well-connected street networks and providing amenities, urban planners and policymakers can promote walkability, sustainable tourism, and the preservation of historic urban areas. This will ultimately contribute to these areas' long-term economic, social, and cultural benefits for tourists and local communities.

The study builds on the existing body of literature that has explored the association between urban layout and walkability, especially in the context of sustainable tourism. By using centrality measures such as closeness, betweenness, and farness, the researchers have added depth to our understanding of how street network configuration impacts

tourist distribution in historical urban areas. In line with previous research, the findings reinforce the significance of betweenness centrality in determining pedestrian movement and visitation patterns, adding to the empirical evidence that street network design can be strategically manipulated to enhance walkability and promote sustainable tourism. The study also offers novel insights by demonstrating that spaces with high betweenness centrality are visited more frequently by tourists, even when they have lower closeness centrality. This suggests a nuanced interaction between different centrality measures and their impact on tourist behavior, contributing to the broader discourse in urban design and planning.

However, it's essential to acknowledge the study's limitations, primarily stemming from the use of limited data. The research primarily focuses on the Turkish part of Nicosia's old city. While the findings are insightful for this specific context, they might only be partially applicable to other historical urban areas with different sociocultural and geographical characteristics. Moreover, the use of student volunteers for data collection could introduce some bias or inconsistency in the results. Additionally, the study relies heavily on centrality values derived from the spatial design network analysis tool (sDNA), which, although powerful, might not capture the full complexity of real-world urban layouts and pedestrian behaviors. These limitations highlight the need for further research incorporating diverse urban settings and more robust data collection methods to validate and extend the findings of this study.

In line with the results, most of the densities on the working days are located in the Bedesten, while the highest density for the off day is situated in the Buyuk Han, although both historic buildings are in the same direction. However, fewer people have been witnessed in the third location, the Venetian Column. Following the city's spatial analysis centrality value, the four locations shared a high centrality value in terms of closeness. Still, they differ in their betweenness, listing the Venetian as the lowest betweenness centrality value, as described in Table 2 below.

**Table 2.** People Density and Layout Centrality Values.

| NO | | IN 29–30 | OUT 29–30 | NQPD | BETWEENNESS |
|---|---|---|---|---|---|
| 1 | Check Point | 70–75 | 73–92 | 119.3 | 0.062 |
| 2 | Buyuk Han | 57–130 | 58–129 | 110.2 | 0.015 |
| 3 | Bedesten | 78–48 | 75–47 | 107.8 | 0.014 |
| 4 | Venetian Column | 36–25 | 37–24 | 109.5 | 0.009 |

To stress more on the discussion, although the Buyuk Han scored a lower value in terms of betweenness and closeness, it showed a higher score in terms of density. This score could refer to the promoted facilities that attract tourists, like local restaurants, coffee shops, and gift and craft shops, while also many live music and local festivals take its place there. These are all included inside the Buyuk Han, as the building structure is surrounded by interior halls and rooms and shaded by an arched walkway with an open-to-sky nave. In comparison, the Bedesten and the Venetian Column lacked such facilities. These findings also reinforce the concept that historical attractions with higher density scores are associated with their ability to attract tourists through their varied amenities. Accordingly, and to strike a balance between historical preservation and tourism development in historic areas, the effect of street networks on the establishment of tourist service facilities must be considered [97].

It is noteworthy to observe a significant correlation between the people density and the centrality values measured for the old city, according to the Freidman two ANOVA test for the everyday people who entered and left the locations with the centrality values in respect of closeness and betweenness, that led to reject the null hypothesis and accept the alternative approach which indicates that the distributions of betweenness, closeness,

Average people entered and left the 32 locations in the old city of Nicosia are of the same values. Table 3 clarifies the result. The spatial analysis of the spatial layout reveals that the checkpoint had the highest centrality values, scoring 119.3 and 0.062 for closeness and betweenness, respectively.

**Table 3.** Freidman's Two Way ANOVA.

| Null Hypothesis | Test | Sig | Decision |
|---|---|---|---|
| The distributions of Betweenness, Closeness, AV-29 and AV-30 are the same | Related-Samples Friedman's Two-Way Analysis of Variance by Ranks | 0.001 | Reject the null hypothesis. |
| Total N | | 32 | |
| Test Statistics | | 81.881 | |
| Degree of Freedom | | 3 | |
| Asymptotic test (two tailed) | | 0.000 | |
| Asymptotic significances are displayed. The significance level is 0.01. | | | |

The previous statement gives a viable explanation for the people's walkability from the checkpoint to the closest point with high values in terms of both closeness and betweenness, which is, in this case, Buyuk Han, having reached the Buyuk Han, the people walk directly to the next station in term of closeness and betweenness. Although the Bedesten has lower closeness values than the Venetian Column, the survey shows that the density in the Bedesten was higher.

## 6. Conclusions

This study contributes to understanding sustainable tourism in historical urban areas by investigating the relationship between street layout centrality and walkability. By examining the impact of betweenness and closeness centrality on tourist distribution and walkability, the research highlights the importance of considering street network configuration when designing urban layouts for sustainable tourism. The findings reveal that betweenness centrality has a more significant influence on tourist distribution than closeness centrality, as tourists visit places with high betweenness centrality more often. This insight is crucial for urban planners and policymakers, as it emphasizes the need to prioritize street networks with high betweenness centrality to promote walkability and enhance the overall tourist experience in historic. Furthermore, the study's methodology offers a valuable approach to examining walkability and tourist distribution in historical urban contexts. By employing manual counts of pedestrian density at intersections and utilizing the spatial design network analysis tool (sDNA) to measure centrality values, the research provides a replicable and adaptable framework that can be applied to other historical urban areas. This contribution is particularly significant for future research exploring the correlation between street layout centrality and factors that potentially influence tourist behavior, such as demographics and attraction types. Moreover, the study's findings can inform the development of specific urban design interventions, such as pedestrianization, that could positively impact both tourist walkability and the centrality values of the street network, ultimately contributing to more sustainable and accessible historical urban spaces.

The research results lead to the conclusion that the street network's centrality values significantly impact the distribution of tourists in historical urban areas, thereby affecting the walkability of these areas. The research showed that the betweenness centrality of the street network has a higher impact on tourist distribution than closeness centrality, as tourists tend to visit places with high betweenness centrality more often. Therefore, it is crucial to consider the street network when designing urban layouts for sustainable tourism. Urban sustainability is a complex concept that encompasses economic growth, social justice, and environmental preservation. Walkability is an essential factor for sustainable urban

planning and plays a crucial role in measuring the sustainability of an urban area. The walkability of an area is related to many indicators that assess and measure urban sustainability. The notion of walkability significantly impacts understanding urban spaces' configurations and, as a result, plays an essential part in evaluating sustainable cities. Therefore, designing walkable urban areas with good connectivity, linkage, paving, and signage is important for promoting sustainable tourism and achieving urban sustainability. This study has also found a clear link between street layout centrality and walkability for sustainable tourism in historic urban areas. Remarkably, the street network's centrality, particularly the betweenness centrality, significantly impacts the distribution of tourists, with areas of high centrality attracting more visitors to them. Therefore, urban planners and policymakers should be encouraged to consider the street network's impact on tourist walkability when designing sustainable and accessible urban spaces. To further develop this study's findings, future work may explore the correlation between street layout centrality and various factors that potentially influence tourist behavior, including tourist demographics and attraction types. Moreover, additional research could investigate the effect of specific urban design interventions, such as pedestrianization, on both tourist walkability and the centrality values of the street network.

**Author Contributions:** Conceptualization, M.A.A. and A.A.; methodology, H.A.N.; software, M.A.A. and A.A.; validation, M.A.A. and A.A.; formal analysis, H.A.N. and M.A.A.; resources, H.A.N.; data curation, H.A.N.; writing—original draft preparation, M.A.A.; writing—review and editing, A.A.; visualization, M.A.A.; supervision, A.A. All authors have read and agreed to the published version of the manuscript.

**Funding:** This research received no external funding.

**Data Availability Statement:** The data will be uploaded to research gate https://www.researchgate.net/profile/Mustafa-Amen (accessed on 20 September 2022).

**Conflicts of Interest:** The authors declare no conflict of interest.

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
