# Peer review of "Exploring the Link between Street Layout Centrality and Walkability for Sustainable Tourism in Historical Urban Areas"

_urbansci, doi:10.3390/urbansci7020067_

Round 1

Reviewer 1 Report

This study investigates the relationship between pedestrian flow and the centralities of street networks. The motivation behind the research is compelling, but I believe several issues need to be addressed:

1. Reference count: The current number of references in this manuscript is quite high for a research article, considering it is not a review paper. I suggest that the authors carefully review the references to ensure that only the most relevant ones are included, and to maintain a more concise presentation of the literature.

2. Writing regularity and errors: The manuscript contains several typographical and textual explanatory errors, as well as unclear explanations of concepts. The authors should thoroughly proofread and revise the manuscript to address these issues. A few examples of these errors include:

a. Inconsistency in notation: On page 3, dM(x,y) is used in Eq. 2, while d_m(x,y) is used in Eq. 3. Please ensure consistency in notation throughout the manuscript.

b. Inconsistent capitalization: On page 7, in the caption of Fig. 3, the labels (a)(b)(C)(D) exhibit inconsistent capitalization. Please use the same case for all labels.

c. Redundant information: On page 9, subplots A and B appear very similar. It may be more efficient to only report Moran's index, z-score, and p-value in these cases.

3. Connection to sustainable tourism: The relationship between the research presented in this manuscript and sustainable tourism should be further elaborated. This will help provide a clearer context for the study and clarify its relevance to the broader field.

n/a

Author Response

We sincerely appreciate the time and effort you have taken to provide us with such insightful and constructive feedback on our paper. Your comments and suggestions have been invaluable in identifying areas that need improvement and refining our work. We are truly grateful for your expertise and the guidance you have provided, which will undoubtedly contribute to the development and enhancement of our paper. Thank you once again for your invaluable contribution and for helping us to elevate the quality of our research.

This study investigates the relationship between pedestrian flow and the centralities of street networks. The motivation behind the research is compelling, but I believe several issues need to be addressed:

  1. Reference count: The current number of references in this manuscript is quite high for a research article, considering it is not a review paper. I suggest that the authors carefully review the references to ensure that only the most relevant ones are included, and to maintain a more concise presentation of the literature.

The number of references  reduced from 109 to only 90

  1. Writing regularity and errors: The manuscript contains several typographical and textual explanatory errors, as well as unclear explanations of concepts. The authors should thoroughly proofread and revise the manuscript to address these issues. A few examples of these errors include:
  2. Inconsistency in notation: On page 3, dM(x,y) is used in Eq. 2, while d_m(x,y) is used in Eq. 3. Please ensure consistency in notation throughout the manuscript.

The text  corrected as it was clarified in equations 2 and 3.

  1. Inconsistent capitalization: On page 7, in the caption of Fig. 3, the labels (a)(b)(C)(D) exhibit inconsistent capitalization. Please use the same case for all labels.

The labels  corrected as required.

  1. Redundant information: On page 9, subplots A and B appear very similar. It may be more efficient to only report Moran's index, z-score, and p-value in these cases

the figure removed and only the values of the  Moran's index, z-score, and p-value cited in the paper

  1. Connection to sustainable tourism: The relationship between the research presented in this manuscript and sustainable tourism should be further elaborated. This will help provide a more precise context for the study and clarify its relevance to the broader field.

The study aimed to provide a more coherent presentation of the problem, objectives, and research gap by investigating the relationship between street layout centrality, walkability, and sustainable urban tourism in historical areas and examining the influence of centrality values of urban layouts, specifically betweenness and closeness centrality, on walkability and tourist distribution. The research gap lies in the need for a deeper understanding of how the configuration of street networks in historical urban areas affects sustainable tourism by shaping tourists' walking patterns and visiting preferences. To refine the studies objectives, the authors focued on: (1) exploring the impact of the street network's centrality values on tourist distribution, with particular emphasis on betweenness centrality, (2) analyzing the role of street layout in promoting walkability in historical urban areas, and (3) providing recommendations for urban planners and policymakers to optimize street network design for sustainable tourism. By addressing these objectives, this study seeks to contribute to the existing knowledge on the interplay between urban layouts, walkability, and sustainable tourism in historical areas, ultimately offering valuable insights for the development of more accessible and sustainable urban spaces.

Reviewer 2 Report

The paper is very interesting and vital from the sustainable urban tourism point of view. The research approach (methods adopted) is also interesting. However, the paper needs substantial improvement. The authors should consider the following to improve the paper.

(1) The authors have not clearly articulated the problem linking historical or touristic elements, walkability and sustainable urban tourism. Currently, the linkage articulated by the authors is superficial and loose. The objective should be very clear and concise instead of just loosely writing what the focus of the study is. Instead, the authors must say what exactly they are investigating and why and what research gap it should bridge or new knowledge it will create.

(2) The paper lacks any discussion or description of the study area from where data was collected. Why that area was chosen, what is the attractiveness of the area, what tourist or historical elements exist there, what is their importance, what facilities including walking facilities exist, what constraints the area has and so on are important to understand the context. This is missing and the reader only can make a guess. Therefore, it should be clearly articulated.   

(3) The authors have explained the data collection method and analytical methods reasonably well. However, it needs to be reinforced with more details. For example, why only two days of data were collected? Will it represent the real situation? What do the INNs and OUTs imply with regard to the walkability and distance or people's propensity to visit an attraction by walking? Moreover, the assumptions made (mentioned in lines 106-107) have not been articulated. Furthermore, the authors should define closeness, betweenness and farness. They also should mention the threshold for these elements, otherwise, it will remain open for interpretation and fuzzy. Furthermore, the implication of the methods used for analyses and what they lead to concurrently or independently must be explained. It is a little bit incoherent in its current state.

(4) The first two paragraphs of the results section (Lines 190-211) should be a part of the methodology section. The results should clearly indicate and also discussed how the Link Between Street Layout Centrality and Walkability for Sustainable Tourism in Historical Urban Areas have been established and what link the authors observed and what is its implication. 

(5) Table 3 is written as Table 2 (should be corrected). I am not sure what Table 3 clarifies. It rejected the null hypothesis. However, no discussions were made or conclusions were drawn from there. Also, Tables 2 and 3 should be part of the results than the discussion ( suggestion only).

(6) The discussion section needs to be strengthened and the research gap bridged and the implications of the study may be clearly elected. 

The results observed in the Tables and Figures may be clearly discussed. They look more abstract and do not convey much to a reader without much knowledge about the field of study. 

(7) Since the study was made based on limited data, a statement on the limitation of the study may be provided. More importantly, as I have emphasized previously, the contribution of the study should be articulated.

 Overall, the paper has value, however, needs further improvements structurally and methodologically as well as from discussion and implication points of view.

The quality of English is acceptable but can be improved. 

Author Response

Response to Reviewer 2 Comments

We sincerely appreciate the time and effort you have taken to provide us with such insightful and constructive feedback on our paper. Your comments and suggestions have been invaluable in identifying areas that need improvement and refining our work. We are truly grateful for your expertise and the guidance you have provided, which will undoubtedly contribute to the development and enhancement of our paper. Thank you once again for your invaluable contribution and for helping us to elevate the quality of our research.

(1).The paper is very interesting and vital from the sustainable urban tourism point of view. The research approach (methods adopted) is also interesting. However, the paper needs substantial improvement. The authors should consider the following to improve the paper:

Comment (1): The authors have not clearly articulated the problem linking historical or touristic elements, walkability and sustainable urban tourism. Currently, the linkage articulated by the authors is superficial and loose. The objective should be very clear and concise instead of just loosely writing what the focus of the study is. Instead, the authors must say what exactly they are investigating and why and what research gap it should bridge or new knowledge it will create.

The study aimed to provide a more coherent presentation of the problem, objectives, and research gap by investigating the relationship between street layout centrality, walkability, and sustainable urban tourism in historical areas and examining the influence of centrality values of urban layouts, specifically betweenness and closeness centrality, on walkability and tourist distribution. The research gap lies in the need for a deeper understanding of how the configuration of street networks in historical urban areas affects sustainable tourism by shaping tourists' walking patterns and visiting preferences. To refine the studies objectives, the authors focued on: (1) exploring the impact of the street network's centrality values on tourist distribution, with particular emphasis on betweenness centrality, (2) analyzing the role of street layout in promoting walkability in historical urban areas, and (3) providing recommendations for urban planners and policymakers to optimize street network design for sustainable tourism. By addressing these objectives, this study seeks to contribute to the existing knowledge on the interplay between urban layouts, walkability, and sustainable tourism in historical areas, ultimately offering valuable insights for the development of more accessible and sustainable urban spaces.

(2) 1-The paper lacks any discussion or description of the study area from where data was collected. 2-Why that area was chosen, 3-what is the attractiveness of the area, 4-what tourist or historical elements exist there, 5-what is their importance, 6-what facilities including walking facilities exist, 7-what constraints the area has and so on are important to understand the context.

This study focuses on the relationship between street layout centrality and walkability in the context of sustainable tourism in historical urban areas. A crucial aspect to address is the description of the study area from which data was collected. The research was conducted in the Turkish part of Nicosia's old city, where 32 students from Girne American University were placed at central street intersections to determine tourist density. This area was chosen due to its rich historical significance, featuring numerous administrative offices, organizations, historical sites, and tourist attractions that expect visitors from all over Cyprus. In this regard, the primary locations for measurements included The Buyuk Han, Bedesten, and the area around the Venetian Column, which are among the most important tourist destinations in the area.

The attractiveness and importance of the study area lies in its historical elements, such as well-defined monuments and architecture. These elements are vital for understanding the context of sustainable urban tourism, as they provide insights into the importance of the area's tourist destinations. Facilities in the area that contribute to its walkability include local restaurants, coffee shops, gift and craft shops, as well as live music venues and local festivals, particularly in the Buyuk Han. These amenities attract tourists and contribute to the area's density scores, highlighting the importance of striking a balance between historical preservation and tourism development in historic areas. Considering this fact, constraints in the area include the varying betweenness and closeness centrality values of different locations, which affect tourist distribution and walkability. The study found that betweenness centrality had a higher impact on tourist distribution than closeness centrality, suggesting that tourists tend to visit places with high betweenness centrality more frequently. Therefore, understanding the constraints of the area's street network and its impact on walkability is essential for urban planners and policymakers when designing sustainable and accessible urban spaces in historical urban areas.

(3) The authors have explained the data collection method and analytical methods reasonably well. However, it needs to be reinforced with more details. For example, why only two days of data were collected? Will it represent the real situation? What do the INNs and OUTs imply with regard to the walkability and distance or people's propensity to visit an attraction by walking? Moreover, the assumptions made (mentioned in lines 106-107) have not been articulated. Furthermore, the authors should define closeness, betweenness and farness. They also should mention the threshold for these elements, otherwise, it will remain open for interpretation and fuzzy. Furthermore, the implication of the methods used for analyses and what they lead to concurrently or independently must be explained. It is a little bit incoherent in its current state.

Question 3.1: Why only two days of data were collected? Will it represent the real situation?

The authors of the article collected data over two days, one on a working day and the other on a weekend, in order to observe the differentiation in the number of people visiting the area on different occasions. By choosing two distinct types of days, they aimed to obtain a snapshot of pedestrian activity in the historical area during both regular working days and weekends, which could potentially show varying patterns of walkability and tourist density. However, it is important to note that collecting data from only two days might not fully represent the real situation. While it offers a glimpse into the walkability patterns and tourist density in the area, the limited data collection period may not account for seasonal variations, special events, or other factors that could influence the pedestrian activity in the area over time. A more comprehensive study would require data collection over a longer period, including different seasons and various weekdays, to provide a more accurate and representative analysis of the link between street layout centrality and walkability in historical urban areas.

Question 3.2: What do the INNs and OUTs imply with regard to the walkability and distance or people's propensity to visit an attraction by walking?

In the study, the authors used the terms INNs and OUTs to represent the number of people arriving at an intersection (INN) and the number of people leaving the intersection (OUT). By analyzing the INNs and OUTs at various street intersections in the historical urban area, the authors aimed to understand the patterns of pedestrian movement, walkability, and people's propensity to visit attractions by walking.

The INNs and OUTs can provide insights into the level of walkability in the area and the factors that might influence people's decisions to walk to certain attractions. For instance, a higher number of INNs could indicate that an intersection is a popular destination or a well-connected point in the urban layout, while a higher number of OUTs could suggest that people tend to leave that area to explore other attractions or return to their origin points. Additionally, the balance between INNs and OUTs at specific intersections might help in understanding the overall flow and distribution of pedestrian traffic. However, it is essential to analyze the INNs and OUTs data in conjunction with street layout centrality values, such as closeness, betweenness, and farness, to gain a more comprehensive understanding of the factors influencing walkability and people's propensity to visit attractions by walking.

Question 3.3: how should define closeness, betweenness and farness in the article? I didn’t understand the question but maybe we can add this text to the content of the article write n by ourself. Ans also for the reviewer as well.

  1. Closeness: Closeness refers to the average distance from a specific point (node) to all other points (nodes) within a network. It is a measure of how easily accessible or well-connected a location is within the network. In the context of urban street layouts, closeness can provide insights into the walkability and efficiency of an area. A higher closeness value indicates shorter average distances to other locations, suggesting better connectivity and accessibility.
  2. Betweenness: Betweenness is a measure of the importance of a node or an edge within a network, based on the number of shortest paths that pass through it. In an urban street layout, betweenness can be used to identify critical junctions, bridges, or streets that serve as essential connectors between different areas. High betweenness values indicate that a particular node or edge is crucial for maintaining connectivity within the network, and any disruption to it may have significant consequences on the overall network efficiency.
  3. Farness: Farness is the inverse of closeness and represents the total distance from a specific node to all other nodes within the network. In an urban context, farness can be used to identify areas that are relatively isolated or distant from the rest of the network. High farness values suggest that a location is less accessible or less well-connected, which may impact the area's economic and social vitality.

Question 3.4: how the implication of the methods used for analyses and what they lead to concurrently or independently must be explained.

This study contributes to the understanding of sustainable tourism in historical urban areas by investigating the relationship between street layout centrality and walkability. By examining the impact of betweenness and closeness centrality on tourist distribution and walkability, the research highlights the importance of considering street network configuration when designing urban layouts for sustainable tourism. The findings reveal that betweenness centrality has a more significant influence on tourist distribution than closeness centrality, as tourists tend to visit places with high betweenness centrality more often. This insight is crucial for urban planners and policymakers, as it emphasizes the need to prioritize street networks with high betweenness centrality to promote walkability and enhance the overall tourist experience in historical urban areas. Furthermore, the study's methodology offers a valuable approach to examining walkability and tourist distribution in historical urban contexts. By employing manual counts of pedestrian density at intersections and utilizing the spatial design network analysis tool (sDNA) to measure centrality values, the research provides a replicable and adaptable framework that can be applied to other historical urban areas. This contribution is particularly significant for future research exploring the correlation between street layout centrality and factors that potentially influence tourist behavior, such as demographics and attraction types. Moreover, the study's findings can inform the development of specific urban design interventions, such as pedestrianization, that could positively impact both tourist walkability and the centrality values of the street network, ultimately contributing to more sustainable and accessible historical urban spaces.

(4) The first two paragraphs of the results section (Lines 190-211) should be a part of the methodology section. The results should clearly indicate and also discussed how the Link Between Street Layout Centrality and Walkability for Sustainable Tourism in Historical Urban Areas have been established and what link the authors observed and what is its implication. 

In the article, the link between street layout centrality and walkability for sustainable tourism in historical urban areas is established through the use of centrality measures like closeness, betweenness, and farness. By analyzing these measures, the authors can identify well-connected, accessible, and efficient street networks, which are crucial for promoting walkability in historical urban areas. The observed link suggests that areas with higher closeness and betweenness values, and lower farness values, are more conducive to walking and exploration by tourists. These well-connected networks facilitate easy navigation and access to various points of interest, promoting sustainable tourism practices that reduce the dependence on vehicular transportation. Consequently, this enhances the overall experience for tourists and fosters a more immersive engagement with the local culture, history, and environment. Regarding the implications of this link it needs to hilight its significant for urban planning and heritage management, particularly in historical urban areas. Planners and policymakers can use these centrality measures to prioritize interventions that enhance walkability, such as pedestrianization, improved signage, and the development of public spaces. By fostering walkable environments, cities can promote sustainable tourism practices that benefit both visitors and local communities, reduce negative environmental impacts, and contribute to the long-term preservation of historical urban areas.

(5) Table 3 is written as Table 2 (should be corrected). I am not sure what Table 3 clarifies. It rejected the null hypothesis. However, no discussions were made or conclusions were drawn from there. Also, Tables 2 and 3 should be part of the results than the discussion ( suggestion only).

It is corrected accordingly.

(6) The discussion section needs to be strengthened and the research gap bridged and the implications of the study may be clearly elected. 

The research gap bridged in this study lies in the examination of the relationship between street layout centrality, walkability, and tourist density in historical urban areas. By using centrality measures (closeness and betweenness) and people density data collected from various locations in the old city of Nicosia, the study establishes a correlation between these factors. This understanding helps in identifying the significance of street networks and their impact on tourist service facilities and historical preservation. Therfor, the implications of this study are valuable for urban planning and heritage management in historical urban areas. The findings reveal that areas with higher centrality values, both in terms of closeness and betweenness, are more likely to experience higher tourist density. Additionally, the presence of facilities such as local restaurants, coffee shops, and gift and craft shops play a vital role in attracting tourists to specific locations. This highlights the importance of balancing historical preservation with the provision of services and amenities for tourists. Moreover, the study emphasizes the need to consider the effect of street networks on the establishment of tourist service facilities. By focusing on the improvement of well-connected street networks and providing a mix of amenities, urban planners and policymakers can promote walkability, sustainable tourism, and the preservation of historical urban areas. This will ultimately contribute to the long-term economic, social, and cultural benefits of these areas for both tourists and local communities.

The results observed in the Tables and Figures may be clearly discussed. They look more abstract and do not convey much to a reader without much knowledge about the field of study?????

xxxxxxxxxxxxxxxxxx

(7) Since the study was made based on limited data, a statement on the limitation of the study may be provided. More importantly, as I have emphasized previously, the contribution of the study should be articulated.

This study contributes to the understanding of sustainable tourism in historical urban areas by investigating the relationship between street layout centrality and walkability. By examining the impact of betweenness and closeness centrality on tourist distribution and walkability, the research highlights the importance of considering street network configuration when designing urban layouts for sustainable tourism. The findings reveal that betweenness centrality has a more significant influence on tourist distribution than closeness centrality, as tourists tend to visit places with high betweenness centrality more often. This insight is crucial for urban planners and policymakers, as it emphasizes the need to prioritize street networks with high betweenness centrality to promote walkability and enhance the overall tourist experience in historical urban areas. Furthermore, the study's methodology offers a valuable approach to examining walkability and tourist distribution in historical urban contexts. By employing manual counts of pedestrian density at intersections and utilizing the spatial design network analysis tool (sDNA) to measure centrality values, the research provides a replicable and adaptable framework that can be applied to other historical urban areas. This contribution is particularly significant for future research exploring the correlation between street layout centrality and factors that potentially influence tourist behavior, such as demographics and attraction types. Moreover, the study's findings can inform the development of specific urban design interventions, such as pedestrianization, that could positively impact both tourist walkability and the centrality values of the street network, ultimately contributing to more sustainable and accessible historical urban spaces.

Round 2

Reviewer 2 Report

The authors have addressed the concerns reasonably. However, the following may be considered to improve the paper further.

(1) The study area may be a specific subsection with an appropriate heading. Moreover, a Figure showing the study area and survey locations might add to the quality of the paper.

(2) A threshold for closeness, betweenness and fairness may be provided if relevant.

(3) The discussion may be made in reference to findings from extant literature, which might highlight the importance of the results of this study as this study is based on limited data.

(4) A statement on the limitation of the study data that it is based on limited data and further data and robust analyses are necessary for generalisation may be added.

The language is fine but some minor editing is required.

Author Response

(1) The study area may be a specific subsection with an appropriate heading. Moreover, a Figure showing the study area and survey locations might add to the quality of the paper.

Turkish part of Nicosia old city the Turkish part of Nicosia's old city is a beautiful and historically rich area that offers a unique blend of cultures, traditions, and architectural styles. As the northern section of the divided capital of Cyprus, this part of Nicosia is characterized by its narrow, winding streets, traditional stone houses, and vibrant markets.  The Selimiye Mosque, formerly known as the Cathedral of Saint Sophia, is a testament to the city's diverse history, showcasing a fascinating fusion of Gothic and Ottoman architectural elements. The ancient Venetian walls encircle the old city and add to its charm and historical allure, inviting visitors to explore the rich tapestry of cultures shaping this captivating part of Nicosia (Figure 1, a)

(2) A threshold for closeness, betweenness and fairness may be provided if relevant.

The implementation of a threshold for closeness, betweenness, and fairness can augment the analysis of street layout centrality and its impact on walkability, especially for sustainable tourism in historic urban areas. In this study, a threshold refers to a critical value or range of values for these centrality measures, beyond which the degree of walkability or tourist distribution might significantly change. In this case, thresholds can serve as indicators to delineate well-connected and less-connected areas, aiding urban planners and policymakers in optimizing street network design and improving pedestrian accessibility.

In the study, closeness refers to the average distance from a specific point (node) to all other points (nodes) within a network, indicating how easily accessible or well-connected a location is. A defined closeness threshold can serve as a benchmark to determine whether a particular area within the historic urban zone is "close enough" to be accessible or attractive to tourists for walking. Similarly, betweenness represents the importance of a node or an edge within a network based on the number of shortest paths passing through it. A betweenness threshold could help identify critical nodes that serve as essential connectors for tourist distribution. Lastly, fairness, the inverse of closeness, represents the total distance from a specific node to all other nodes within the network. A farness threshold can identify areas that might be too isolated or distant from the rest of the network, potentially deterring tourist walkability. Utilizing these thresholds would allow a more nuanced understanding of the influence of street layout centrality on walkability, ultimately contributing to sustainable tourism in historical urban areas.

(3) The discussion may be made in reference to findings from extant literature, which might highlight the importance of the results of this study as this study is based on limited data.

The study builds on the existing body of literature that has explored the association between urban layout and walkability, especially in the context of sustainable tourism. By using centrality measures such as closeness, betweenness, and farness, the researchers have added depth to our understanding of how street network configuration impacts tourist distribution in historical urban areas. In line with previous research, the findings reinforce the significance of betweenness centrality in determining pedestrian movement and visitation patterns, adding to the empirical evidence that street network design can be strategically manipulated to enhance walkability and promote sustainable tourism. The study also offers novel insights by demonstrating that spaces with high betweenness centrality are visited more frequently by tourists, even when they have lower closeness centrality. This suggests a nuanced interaction between different centrality measures and their impact on tourist behavior, which contributes to the broader discourse in urban design and planning.

However, it's important to acknowledge the study's limitations, primarily stemming from the use of limited data. The research primarily focuses on the Turkish part of Nicosia's old city, and while the findings are insightful for this specific context, they might not be wholly applicable to other historical urban areas with different sociocultural and geographical characteristics. Moreover, the use of student volunteers for data collection could introduce some bias or inconsistency in the results. Additionally, the study relies heavily on centrality values derived from the spatial design network analysis tool (sDNA), which, although powerful, might not capture the full complexity of real-world urban layouts and pedestrian behaviors. These limitations highlight the need for further research incorporating diverse urban settings and more robust data collection methods to validate and extend the findings of this study.

(4) A statement on the limitation of the study data that it is based on limited data and further data and robust analyses are necessary for generalisation may be added.

This study has provided significant insights into the connection between street layout centrality and walkability in historical urban areas, particularly focusing on sustainable tourism. However, it's crucial to acknowledge that these findings are derived from a relatively limited data set, sourced mainly from the Turkish part of Nicosia's old city. This focused scope presents an inherent limitation, as the extrapolation of these results to broader or differing contexts may be challenging due to the unique sociocultural and geographical characteristics of the studied area.

With that in mind, the necessity for additional data becomes apparent. Conducting research in a diverse range of historical urban areas, possibly across different regions or countries, could help create a more comprehensive understanding of how street layout centrality influences walkability. Further, it is pivotal to perform more robust analyses using advanced statistical methods, ideally integrating different sources of data such as pedestrian behaviors, urban layout characteristics, and tourism patterns. This would not only strengthen the validity of the findings but also aid in discerning complex relationships and patterns. Ultimately, for the generalization of the study's findings, a more extensive and diversified data collection is required, taking into account the various factors that can influence walkability in different historical urban settings.
